# Willingness to Oppose Smoking among Pregnant Women

**DOI:** 10.3390/ijerph182111636

**Published:** 2021-11-05

**Authors:** Dominik Olejniczak, Krzysztof Klimiuk, Urszula Religioni, Anna Staniszewska, Mariusz Panczyk, Agnieszka Nowacka, Paulina Mularczyk-Tomczewska, Edyta Krzych-Fałta, Anna Korcala-Wichary, Łukasz Balwicki

**Affiliations:** 1Department of Public Health, Faculty of Health Science, Medical University of Warsaw, 02-091 Warsaw, Poland; dominik.olejniczak@wum.edu.pl (D.O.); paulina.mularczyk@wum.edu.pl (P.M.-T.); 2Faculty of Medicine, Medical University of Gdańsk, 80-210 Gdańsk, Poland; 3Collegium of Business Administration, Warsaw School of Economics, 02-554 Warsaw, Poland; urszula.religioni@gmail.com; 4Department of Experimental and Clinical Pharmacology, Medical University of Warsaw, 02-091 Warsaw, Poland; anna.staniszewska@wum.edu.pl; 5Department of Education and Research of Health Sciences, Faculty of Health Sciences, Medical University of Warsaw, Zwirki Wigury 81, 01-580 Warsaw, Poland; mariusz.panczyk@wum.edu.pl; 6Department of Obstetrics and Gynecology Didactics, Faculty of Health Sciences, Medical University of Warsaw, 02-091 Warsaw, Poland; agnieszka.nowacka@wum.edu.pl; 7Nursing Basics Facility, Medical University of Warsaw, 02-091 Warsaw, Poland; edyta.krzych-falta@wum.edu.pl; 8Health Promotion Section, Students’ Scientific Public Health Association, Medical University of Warsaw, 02-091 Warsaw, Poland; anna.korcala@gmail.com; 9Department of Public Health and Social Medicine, Medical University of Gdańsk, 80-210 Gdańsk, Poland; balwicki@gumed.edu.pl

**Keywords:** smoking, relationship, prevention, pregnancy, tobacco, reprimand

## Abstract

Even though smoking causes numerous threats to the developing foetus, it is the most common addiction in Polish women of reproductive age. Most studies undertake to examine the subject of opposing second-hand smoking or creating tools to reprimand pregnant women more effectively using a qualitative methodology. The study aimed to determine the profile of a pregnant woman who is willing to oppose the smoking of another pregnant woman. The research was conducted using an original multiple-choice questionnaire. The survey was shared on websites for expecting parents. Completed questionnaires were collected from 11,448 pregnant women. The Wald test for logistic regression was used for statistical analysis. Predictors of whether someone would draw another pregnant women’s attention to the fact that smoking is harmful were: intellectual work (OR 1.136; *p*-value 0.020) and currently being a student (OR 1.363; *p*-value 0.004), involvement of the child’s father (OR 1.377; *p*-value < 0.001), contact with social campaigns (OR 1.150; *p*-value 0.005) and knowledge about the consequences of smoking, as well as talking to the midwife about the harmfulness of cigarettes during pregnancy (OR 1.655; *p*-value < 0.001). Interpersonal relationships leave scope for public health interventions. It is worth enhancing criticism against smoking by specialists through information and education campaigns.

## 1. Introduction

Even though various studies have shown that smoking can cause numerous threats to the developing foetus [1,2], it is nevertheless the most common addiction in Polish women of reproductive age [3]. Possible complications include low birth weight, congenital abnormalities, neonatal or sudden infant death syndrome or even an ectopic pregnancy [1]. The problem of smoking among pregnant women still needs to be resolved. The data show that up to 25% of women smoked at some point during pregnancy, and a significant proportion of women were exposed to second-hand smoke [4,5].

According to the International Federation of Gynaecology and Obstetrics, it is recommended even for women only attempting to become pregnant to cease smoking [6]. The conducted studies show that quitting smoking, even in the third to fourth month of pregnancy, reduces the chances of many complications, such as premature birth [1]. Moreover, it is more manageable for parents to quit smoking during pregnancy [7,8], although the effect of this decision is not always permanent [8]. The instability of this decision can be related to the influence of smokers in the environment of the pregnant women [9]. Therefore, understanding interpersonal relations and interventions can help to strengthen the effect of quitting smoking in pregnancy.

Substantial research has focused on the pressures exerted on pregnant women to protect the health of their foetus and on the importance of these demands for quitting smoking permanently [10,11,12,13]. The pressures can be increased by cultural or religious norms in communities [14] or the sense of responsibility felt by an individual family member for the immediate situation [15]. Moreover, it can be observed that not only the pregnant person, but also the environment is affected, as it is easier for other people to reprimand a smoker when the victim of passive smoking is a mother-to-be [16].

Although it has been confirmed that pregnant women feel an intense conflict when someone smokes in their surroundings [17], little research has been done to assess the relations and support in an exclusive group of pregnant women. Most studies undertake to examine the subject of opposing second-hand smoking [17,18] or creating tools to reprimand pregnant women more effectively [15] using a focus methodology. The aim of our study was to determine the profile of a pregnant woman who is willing to oppose the smoking of another pregnant woman.

## 2. Materials and Methods

### 2.1. Data Collection

The research was conducted from November 2016 to April 2017 using an original questionnaire specifically designed for the purpose of the study. Computer-assisted web interviewing (CAWI) was used to reach the respondents. The snowball sampling method was used to collect the questionnaires, i.e., a technique in which respondents recruit more people online to participate in a study. Thus, it was possible to locate the hidden population, expand the number of completed questionnaires and improve the study’s reliability. The survey was shared on Polish websites for expecting parents (https://www.babyboom.pl/ (accessed on 1 November 2021) and https://forum.parenting.pl/ (accessed on 1 November 2021)) with a request for completion that was addressed at pregnant women, regardless of pregnancy trimester. Respondents were informed that the survey was for research purposes, and they were notified that their responses would remain anonymous. It was a condition for submitting the questionnaire to answer each question in order to exclude incomplete data from the analysis.

### 2.2. Questionnaire

The original multiple-choice questionnaire consisted of 41 questions. It included variables about socioeconomic data, information about the current pregnancy, medical care, welfare, smoking during pregnancy, awareness of the harmfulness of smoking, exposure to passive smoking and subjective assessment of one’s own smoking. Willingness to oppose the smoking of another pregnant woman was identified using the question: Would you draw the attention of a pregnant woman who was a stranger to the fact that smoking is harmful? The Medical University of Warsaw gave ethical consent to run the study. The questionnaire was created using Google Forms [19] and consisted of 41 questions.

### 2.3. Statistics

The logistic regression model was used to determine the probability of a pregnant woman reprimanding a smoking person. For each of the predictors, which were variables used in the survey, the following rates were determined: beta coefficient, odds ratio, confidence interval, Wald test and probability value.

## 3. Results

Completed questionnaires were collected from 11,448 pregnant women. The sociodemographic information about the participants is presented in Table 1. Of the participants in the study, 80.5% reported that their pregnancy was a planned one. In addition, 38.1% of respondents smoked before pregnancy, and only 11.1% did it during pregnancy.

Of the participants in the study, 75.91% indicated that they had not been advised by a gynaecologist on the harmful effects of smoking. This result differs between current smokers and non-smokers: 60.58% and 77.82%, respectively (*p* < 0.001).

However, 51.7% of respondents had come across an anti-smoking campaign. When asked if they would draw a pregnant woman’s attention to the harmfulness of smoking, 87.9% responded positively, saying that they would reprimand a family member, 87.7% would do this if the smoker was a close friend, 58.3% would react in the presence of a distant friend and 19.4% would address a stranger. A statistical overview of the factors influencing respondents’ reactions is presented in Table 2.

## 4. Discussion

In this study, we found certain factors which may affect the probability of reprimanding smoking in pregnancy by another pregnant woman. The obtained results imply a profile of a well-educated woman with a high level of awareness and emotional support. This profile may help reduce the number of individuals who are the most vulnerable to second-hand smoke. It also allows us to determine factors that increase pregnant women’s criticism to smoke, which is essential for introducing effective interventions [20].

Various studies have repeatedly indicated personal interactions and relationships as factors helping two partners to achieve smoking cessation [21] and to stay abstinent [22]. Moreover, pregnancy is a period that significantly changes couples’ dynamics, being a specific occasion when couples decide to quit smoking—with diverse effects [18,23]. Similarly, a partner’s support when quitting smoking is indispensable [18,24,25].

The conducted research shows that knowledge of the consequences of smoking and FTS increases the probability of reprimanding other smoking women. This can be related to the results obtained in other studies that have shown that women familiar with the consequences of smoking quit smoking more efficiently [26,27] and that the knowledge of the negative consequences of smoking for the foetus is needed to quit [28]. Moreover, many studies indicate that women with higher education smoke less and quit smoking during pregnancy more often [29,30], which can also be associated with the results obtained in this study, where women who were still students during pregnancy were more critical of smoking. As in the studies on smoking itself, these results are linked to data on work life. Women doing intellectual work are much more likely to reprimand other smokers. This is reflected in the fact that these women smoke less and have a greater chance of breaking the addiction permanently [26,29]. It may be related to job requirements, such as education mentioned earlier, or to a higher socioeconomic level.

In light of the above, it is important to raise awareness of the adverse effects of smoking, e.g., through information campaigns. Even though campaigns aimed at fighting tobacco smoking are not always as effective as we would like them to be [31], they still have a positive effect, and it is worth implementing them [32,33]. For this reason, it is crucial to implement an appropriate campaign to raise awareness of the effects of smoking [34] and to target the social determinants of health through poverty reduction, housing and educational support [35]. Additionally, as our study and others show [36,37,38], midwives play an important role in education about smoking and its negative impact on health. However, such an effect was not observed in the case of gynaecologists caring for women in our study.

This difference may be because midwives are a vital group of health professionals who can influence pregnant women [37]. It can be observed that the lack of doctors’ support is sometimes the main barrier to smoking cessation [39]. It is important since doctors have a significant influence on patients [40] because they are considered reliable [40]. Physicians indicate the lack of visible effects [41] and the lack of time [42] as a barrier to action. The solution to such problems may be providing doctors with clear guidelines on what to convey to the patient [43].

Another factor that negatively affects giving reprimands to others is when a woman still smokes while pregnant. Additionally, it is statistically significant that the chance of criticising others is reduced by being exposed to second-hand smoking only once a month. It is known that women in whose household someone smokes regularly are less likely to quit [26,35]. A significant factor associated with smoking cessation is low exposure to second-hand smoking [29]. This shows that our criticism towards smoking decreases the more we are exposed to cigarette smoke.

Interestingly and statistically significantly, women in their third pregnancy (not earlier or later) are more likely to reprimand another person. This is probably due to a complex sociodemographic cause, and the result is the effect of correlation. Although contrary to the outcome obtained, many studies indicate that having children from earlier pregnancies is a stressor that causes failure to quit smoking during pregnancy [26,44]. Moreover, pregnancy itself is associated with many stresses [29,45], which may not affect the criticism of smoking so much, but the reluctance to interfere with other people’s behaviour.

The results obtained can be helpful in the development of campaigns to promote smoking cessation. However, it is essential not to simplify these results, as this field is not fully understood by pregnant women, and excessive generalisation may lead to stereotypes [46]. Many studies show that stigma-centred campaigns can be ineffective or even discouraging [47,48], whereas some qualitative studies show that pregnant women prefer positive and empowering smoking cessation ads [49]. Therefore, conclusions need to be drawn carefully.

As the authors, we are aware of the limitations of the conducted research. First of all, the research was conducted with the use of a proprietary questionnaire designed for the purpose of the research. Despite making every effort and consulting with experts for the questionnaire, the research carried out with the use of this tool is difficult to compare with other studies in this area. Additionally, in the designed survey, we did not ask for other information that could affect the prevalence of smoking among women, which may, to some extent, interfere with the results of the study.

A characteristic limitation of a study using the CAWI method is that only respondents with access to a specific online platform can complete the survey. Participants in our study were recruited from particular forums, which reduces the population’s representativeness and the impact of the study. Furthermore, the snowball sampling technique indicates the notion of non-randomisation [46]. The questionnaire surveys also fail to objectively verify women’s smoking status while pregnant, as it has been pointed out that this information is underestimated in the case of self-reporting [50]. Due to the fact that the socioeconomic and cultural situation of pregnant women differs from country to country, the results may not be transferable to other cultures in the world or even in Europe.

## 5. Conclusions

Despite the lack of a random group, it is one of the largest studies exploring the subject of interpersonal interventions in smoking in terms of a sample of a population. The conducted research suggests the presence of certain trends. It can be noted that actions aimed at raising awareness of the harmfulness of smoking in pregnancy increase the chance of taking an action that criticises smoking. Together with other outcomes, such as a higher level of education and partner’s support, this shows us a pattern where we can crystallise the figure of a person who is more inclined to interpersonal intervention.

Interpersonal relationships leave scope for public health interventions. It is worth enhancing criticism against smoking by specialists through information and education campaigns, as shown in the conducted research. However, there are still opportunities to enhance this type of activity, such as increasing the number of gynaecologists reporting on the effects of smoking. It seems promising to explore factors influencing smoking across society, as knowledge of the mechanism favouring smoking could further reduce the aforementioned behaviour.

## Figures and Tables

**Table 1 ijerph-18-11636-t001:** Characteristics of participants in the study (n = 11,448).

	N	%
**Age Group**		
14–17	12	0.1
18–20	277	2.4
21–25	2942	25.7
26–30	5709	49.9
31–40	2472	21.6
40 and over	36	0.3
**Education**		
Primary	35	0.3
Junior secondary	110	1.0
Vocational	460	4.0
Secondary	2948	25.8
Higher non-medical	6824	59.6
Higher medical	1071	9.4
**Place of Residence**		
Countryside	2666	23.3
Town of up to 50 thousand residents	2422	21.1
Town of 51–200 thousand residents	2252	19.7
City of 201–500 thousand residents	1361	11.9
City of more than 500 thousand residents	2747	24.0
**Work Life**		
I’m a student/university student	669	5.8
I work and study	351	3.1
I’m on sick leave	6190	54.1
I work	2685	23.5
I don’t work	1553	13.6
**Marital Status**		
Informal relationship	2937	25.7
Married	8403	73.4
Single mother	91	0.7
Separated/divorced	17	0.1
**Number of Pregnancies**		
1	7194	62.8
2	3228	28.2
3	794	6.9
≥4	232	2.0
**Multiplicity of Pregnancy**		
Single pregnancy	10,848	94.8
Multiple pregnancy	431	3.8
No data	169	1.5

**Table 2 ijerph-18-11636-t002:** Model of logistic regression for the “yes” variable with regard to the question “Would you draw the attention of a pregnant woman who was a stranger to the fact that smoking is harmful?”

Predictor	Level Effect	B	OR	−95% CI	+95% CI	t Wald	*p*-Value
Independent part		−2.134	0.118	0.051	0.276	24.372	<0.001
1. Which pregnancy is it?	First (ref.)						
	Second	−0.081	0.922	0.822	1.034	1.915	0.166
	Third	0.236	1.267	1.053	1.524	6.310	0.012
	Fourth or more	0.024	1.025	0.729	1.441	0.020	0.889
2. Which week of pregnancy is it?	1–13 week (ref.)						
	14–26 week	−0.165	0.848	0.744	0.966	6.123	0.013
	Over 26 weeks	−0.225	0.798	0.698	0.914	10.694	0.001
4. Are you under constant care of a gynaecologist during pregnancy?		0.190	1.210	0.810	1.807	0.866	0.352
5. This pregnancy was	Not planned (ref.)						
	Planned	−0.087	0.917	0.807	1.041	1.804	0.179
7. What is your professional situation?	I work (ref.)						
	I don’t work	0.060	1.062	0.896	1.257	0.479	0.489
	I study	0.310	1.363	1.105	1.681	8.397	0.004
	I work and study	0.077	1.080	0.820	1.422	0.301	0.583
	I’m employed and on sick leave	−0.014	0.986	0.871	1.117	0.049	0.826
8. What is your job?	Physical work (ref.)						
	Intellectual work	0.128	1.136	1.020	1.266	5.418	0.020
12. How could you describe the involvement of your child’s father in the preparation for the child’s birth?	No or little involvement (ref.)						
	High involvement	0.320	1.377	1.217	1.558	25.764	<0.001
20. Have you smoked during pregnancy? (since the moment you learnt about the pregnancy)		−0.436	0.647	0.535	0.782	20.138	<0.001
24. Has the gynaecologist or midwife talked to you about the harmfulness of smoking during pregnancy?		0.504	1.655	1.487	1.842	85.079	<0.001
25. Are you exposed to second-hand smoking? (inhaling smoke of cigarettes smoked by others in your presence)	No (ref.)						
	Yes, every day	0.033	1.034	0.876	1.221	0.156	0.693
	Yes, a few times a week	−0.096	0.908	0.764	1.080	1.187	0.276
	Yes, a few times a month	−0.148	0.862	0.741	1.003	3.682	0.055
	Yes, but less than once a month	−0.369	0.692	0.604	0.792	28.343	<0.001
29. Have any of your friends/family members smoked cigarettes during pregnancy?		−0.213	0.808	0.731	0.893	17.348	<0.001
32. Do you think smoking cigarettes during pregnancy can adversely affect the health of the unborn child?		0.864	2.372	1.321	4.259	8.367	0.004
35. Have you ever heard of FTS? (foetal tobacco syndrome)		0.501	1.651	1.496	1.822	99.071	<0.001
36. To what extent do you agree with this statement? “Health behaviour of a pregnant woman has little impact on the child’s health.”	I do not agree (ref.)						
	Hard to say	−0.383	0.682	0.455	1.021	3.454	0.063
	I agree	−0.006	0.994	0.895	1.104	0.013	0.910
37. Have you ever come across an information campaign about the harmfulness of smoking during pregnancy?		0.140	1.150	1.044	1.268	7.983	0.005

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
