# Peer review of "Willingness to Oppose Smoking among Pregnant Women"

_ijerph, 2021, doi:10.3390/ijerph182111636_

Round 1

Reviewer 1 Report

This is a manuscript on a questionnaire on attitudes of willingness to oppose smoking among pregnant women. The authors have used one statistical method repeatedly with no stratification on other data. No multivariable testing was used. 

Major comments

In the abstract mental work is mentioned with on line 31 an odds ratio of 1,1136. Neither of these is found in the tables. Instead, Intellectual work has an OR of 1,136 in table 2.  

Also, on line 31 the term “study” is incomprehensible. Maybe change to “currently being a student” or something else which explains this term better.

Reading the results of table 2 and the results in the discussion, some parts of the discussion is contradicted by table 2 or not supported by data.

Question 10 on the relationship with the father. It is not appropriate to use no relation as a reference as this group seems to be the single mother group of 91 subjects; in contrast, married is 8403, and informal is 2937. Also, the group of single mothers may differ in many other ways. Age may be one of these differences.

Secondly, the statement of a difference in attitude in the discussion on line 134 between married and unmarried is not true. Using the small reference group mentioned above, both these have overlapping confidence intervals, clearly speaking against this statement. To conclude a difference, a statistical analysis is needed comparing married and unmarried.

In discussion lines 185-190, the authors state that being in a third pregnancy makes a woman less likely to reprimand another person and speculate on reasons for this. Looking in table 2, question 1 shows an OR of 1.267, showing instead that they are more likely to reprimand other people. Thus there are discrepancies showing that this part of the discussion is not supported by data.

The discussion is very long and must be rewritten in a shorter, less speculative manner with an accurate interpretation of the data presented. It also needs to be more humble as the results may not be transferable to other cultures in the world or not even to neighbors of Poland.

The statement in conclusion that “The conducted research shows that actions aimed at raising awareness of the harmfulness 208 of smoking in pregnancy have a positive effect.” is not supported by data but instead may represent the authors' personal opinions independent of the study results.

Minor comments.

In the abstract, please insert a blank between OR and the numbers.

In the abstract line 31: The term “mental work”

In the ethics part lines 91-92, please insert the administrative number of the ethical approval for traceability.

Results, line 108: 77.8% had been advised by a gynecologist on the adverse effects of smoking. Did this frequency differ between current smokers and non-smokers?

Table 2 heading  ”pregna/moking” apparently misspelled

DO not use “,” as a decimal separator in table 2. Change to “.”

Please change p-values of 0.000 to <0.001

An English version of a blank questionnaire should be provided as supplementary material, as not all questions can be seen in the manuscript. This version must be professionally translated to English.

Author Response

October 15th, 2021

Dear Reviewer,

We would like to thank you very much for the valuable remarks concerning the article entitled “Willingness to oppose smoking among pregnant women”. We have followed every recommendation of reviewers. We marked all changes in the text in the track changes mode.

The answers to individual questions of the reviewer and the detailed changes made to the manuscript are presented below:

Review Report (Reviewer 1)

Comments and Suggestions for Authors

This is a manuscript on a questionnaire on attitudes of willingness to oppose smoking among pregnant women. The authors have used one statistical method repeatedly with no stratification on other data. No multivariable testing was used.

Thank you very much for your review. We have addressed each issue below. We hope that thanks to this we have significantly improved the quality of the article.

Major comments

  1. In the abstract mental work is mentioned with on line 31 an odds ratio of 1,1136. Neither of these is found in the tables. Instead, Intellectual work has an OR of 1,136 in table 2.

Thank you very much for this suggestion. To make things more consistent, we changed "mental work" to "intellectual work". Additionally, the entry "1.1136" has been corrected to "1.136".

  1. Also, on line 31 the term “study” is incomprehensible. Maybe change to “currently being a student” or something else which explains this term better.

Thank you very much, we have changed these phrases as suggested by the reviewer.

  1. Reading the results of table 2 and the results in the discussion, some parts of the discussion is contradicted by table 2 or not supported by data.

The authors reread and largely redrafted the discussion. We tried to take into account all comments from reviewers regarding data analysis and their discussion with the results of other authors. We believe these parts of the work are now more consistent.

  1. Question 10 on the relationship with the father. It is not appropriate to use no relation as a reference as this group seems to be the single mother group of 91 subjects; in contrast, married is 8403, and informal is 2937. Also, the group of single mothers may differ in many other ways. Age may be one of these differences.

Thank you very much for this suggestion. Indeed, this issue can be confusing. Therefore, at the moment, we have decided to remove these elements from the analysis so as not to mislead our readers. If the reviewer recognizes that the group "no relation" cannot be the reference group, we can try to do a new analysis, which, however, may significantly affect the results.

  1. Secondly, the statement of a difference in attitude in the discussion on line 134 between married and unmarried is not true. Using the small reference group mentioned above, both these have overlapping confidence intervals, clearly speaking against this statement. To conclude a difference, a statistical analysis is needed comparing married and unmarried.

After re-analyzing the results and discussing, we fully agree with the reviewer. We made the necessary corrections to make the text unambiguous.

  1. In discussion lines 185-190, the authors state that being in a third pregnancy makes a woman less likely to reprimand another person and speculate on reasons for this. Looking in table 2, question 1 shows an OR of 1.267, showing instead that they are more likely to reprimand other people. Thus there are discrepancies showing that this part of the discussion is not supported by data.

Thank you very much for this suggestion. This part of the work has been modified according to the reviewer's indicators.

  1. The discussion is very long and must be rewritten in a shorter, less speculative manner with an accurate interpretation of the data presented. It also needs to be more humble as the results may not be transferable to other cultures in the world or not even to neighbors of Poland.

Of course, we have edited the discussion as directed. We also described the limitations of the study. Thank you very much for the reviewer's suggestions.

  1. The statement in conclusion that “The conducted research shows that actions aimed at raising awareness of the harmfulness 208 of smoking in pregnancy have a positive effect.” is not supported by data but instead may represent the authors' personal opinions independent of the study results.

The study conclusion has been edited to refer to the results.

Minor comments.

  1. In the abstract, please insert a blank between OR and the numbers.

Thank you for this suggestion. Blank between OR and the numbers have been applied.

  1. In the abstract line 31: The term “mental work”

As suggested by the reviewer, we changed mental work "to" intellectual work ".

  1. In the ethics part lines 91-92, please insert the administrative number of the ethical approval for traceability.

Thank you for this suggestion. Information on the consent of the bioethical commission is included at the end of the article, because in Poland, such consent is not required in the survey.

  1. Results, line 108: 77.8% had been advised by a gynecologist on the adverse effects of smoking. Did this frequency differ between current smokers and non-smokers?

Yes, we have conducted an additional analysis which shows that these results vary (p<0.001). We wrote this information in the results lines 114-115.

  1. Table 2 heading ”pregna/moking” apparently misspelled

We apologize for this oversight. Of course, we improved the words.

  1. DO not use “,” as a decimal separator in table 2. Change to “.”

As the reviewer suggests, we have changed the decimal separator.

  1. Please change p-values of 0.000 to <0.001

Thank you, the change has been applied.

  1. An English version of a blank questionnaire should be provided as supplementary material, as not all questions can be seen in the manuscript. This version must be professionally translated to English.

Of course, the English version of the questionnaire will be sent as a supplementary file.

Once again, thank you very much for your comments.

Yours faithfully, 

Authors

Reviewer 2 Report

REVIEW nr. 1380568

The subject covered by the authors, although interesting, has already been extensively explored in previous works. This is essentially a frequentist approach in support of existing literature. The debate seems to have been dealt with in a little detail. The type of research and the consistency of the sample coincide with the purposes of the journal, finally, although not native language, I think that English should be revised better.

My remarks:

###  1. Abstract: “…..was conducted using an original questionnaire”. Specify the type of questionnaire in the abstract. See also subsequent comments on the questionnaire.

### 2. Abstract: Insert in the abstract the statistical methodology used: logistic model/Wald approach.

### 3. Abstract: Insert the p-values in the corresponding OR.

### The authord stated (lines 72-73: “The cross-cutting research was conducted from November 2016 to April 2017 using an original questionnaire…..”. Support or specify better.

###  2.2  Questionnaire: is this a validated questionnaire? (see reference nr. 19) Specify, there is no detailed and more in-depth information on the subject. Has it been taken up by google and reworked by the authors? With what criteria? Deepen this aspect in a substantial way. If not, do the authors intend to use the questionnaire for possible validation?

### The questionnaire lacks, for example, information on possible past pathologies of the participants, a possible confounding factor in the results presented in Table 2.

###  Results: Curious the treatment of the results, in practice, we read two Tables (1 and 2) without a minimum of reasoned decomposition of the results themselves. It is not a critical remark in particular, but an invitation to review the presentation of the results in a different way.

### Table 2: Was the choice of the reference variable (ref) rationally identified by the authors? Which statistical software was used? Specify.

### Discussion: there is a lack of substantive research findings, especially in relation to the references mentioned. I suggest revising this part of the manuscript by articulating better what emerged from the results. Highlight the novelty aspects of the work; what does the research add again?

### Better clarify sample selection procedures. Also, does the sample refer to Poland? Insert into materials and methods?

### Limitations of the study? What?

Author Response

October 15th, 2021

Dear Reviewer,

We would like to thank you very much for the valuable remarks concerning the article entitled “Willingness to oppose smoking among pregnant women”. We have followed every recommendation of reviewers. We marked all changes in the text in the track changes mode.

The answers to individual questions of the reviewer and the detailed changes made to the manuscript are presented below:

Review Report (Reviewer 2)

REVIEW nr. 1380568

The subject covered by the authors, although interesting, has already been extensively explored in previous works. This is essentially a frequentist approach in support of existing literature. The debate seems to have been dealt with in a little detail. The type of research and the consistency of the sample coincide with the purposes of the journal, finally, although not native language, I think that English should be revised better.

Thank you very much for your review and appreciation of our work. We undertook this area of ​​research because it is not yet well known in Poland. Below we have answered all the reviewer's suggestions, for which we would like to thank you. Additionally, we made a linguistic correction. We hope the text is now much more relevant.

My remarks:

  1. ### 1. Abstract: “…..was conducted using an original questionnaire”. Specify the type of questionnaire in the abstract. See also subsequent comments on the questionnaire.

Thank you for this suggestion. Information has been added to abstract and to the methods section (multiple choice questionnaire).

  1. ### 2. Abstract: Insert in the abstract the statistical methodology used: logistic model/Wald approach.

Information has been added.

  1. ### 3. Abstract: Insert the p-values in the corresponding OR.

The p-value values ​​have been included in the abstract.

  1. ### The authord stated (lines 72-73: “The cross-cutting research was conducted from November 2016 to April 2017 using an original questionnaire…..”. Support or specify better.

After careful analysis and discussion between the authors, we decided to delete the cross-cutting research entry as it is not appropriate. We apologize for this imperfection.

  1. ### 2.2 Questionnaire: is this a validated questionnaire? (see reference nr. 19) Specify, there is no detailed and more in-depth information on the subject. Has it been taken up by google and reworked by the authors? With what criteria? Deepen this aspect in a substantial way. If not, do the authors intend to use the questionnaire for possible validation?

The survey was designed specifically for the purpose of the study, based on a previous literature review. The questionnaire was consulted statistically and with 2 experts in the scope of the study. Then, we piloted the survey on 10 potential respondents. The corrected survey was distributed online. We have described this information in the methods section and also at the end of the discussion in limitations, as we are aware of the limitations of the use of this tool.

  1. ### The questionnaire lacks, for example, information on possible past pathologies of the participants, a possible confounding factor in the results presented in Table 2.

Thank you very much for this suggestion. Indeed, there are factors that may have influenced smoking prevalence that we did not ask about in the survey. This is a very important limitation that we described at the end of the manuscript.

  1. ### Results: Curious the treatment of the results, in practice, we read two Tables (1 and 2) without a minimum of reasoned decomposition of the results themselves. It is not a critical remark in particular, but an invitation to review the presentation of the results in a different way.

After the discussion between the authors, it seems to us that the presentation of data in tables is the most legible. For this reason, we have decided to leave this way of presenting the results. Of course, some data can be presented in the form of graphs, but there would be quite a lot of graphs, which in our opinion will reduce the readability of the results. However, if the reviewer thinks otherwise, we will try to present some of the results in a graphical form.

  1. ### Table 2: Was the choice of the reference variable (ref) rationally identified by the authors? Which statistical software was used? Specify.

The reference variables were selected on the basis of an initial literature review, at the stage of designing the study and preparing the survey.

All calculations were performed with STATISTICATM 13.3 software (TIBCO Software, Palo Alto, California, United States). For all analyses, the P-level of < 0.05 was considered statistically significant.

  1. ### Discussion: there is a lack of substantive research findings, especially in relation to the references mentioned. I suggest revising this part of the manuscript by articulating better what emerged from the results. Highlight the novelty aspects of the work; what does the research add again?

The conclusions on the basis of the obtained results as well as the novelty of the work were emphasized in the conclusion.

  1. ### Better clarify sample selection procedures. Also, does the sample refer to Poland? Insert into materials and methods?

In our manuscript, sample selection is described as snow ball. The sample of the population is large, but it was not selected representative. The survey was published only in Polish. We described all the elements that could affect the quality of the study as limitations.

  1. ### Limitations of the study? What?

The last part of the discussion presents the limitations that result from the conducted research.

Once again, thank you very much for your comments.

Yours faithfully, 

Authors

Round 2

Reviewer 1 Report

Thank you for the new version of which has improved significantly.

I have some questions on the percentages, lines 110-112

"77.8% of the participants to the study indicated that they had not been advised by a gynecologist on the harmful effects of smoking. This result differs between current smokers and non-smokers: 60.58% and 77.82%, respectively (p<0.001)."

If the results differ between 60.58% and 77.82% - first line "77.8% of the participants" must be incorrect. In all participants, this percentage must be somewhere between 60-77%, i.e. between smokers and non-smokers.

Line 110 "to the study" should be "in the study"

Author Response

October 21th, 2021

Dear Reviewers,

We would like to once again thank you very much for the remarks concerning our article entitled “Willingness to oppose smoking among pregnant women”. We marked all changes in the text in the track changes mode.

The answer to the second review report and changes made to the manuscript are presented below.

Review Report (Reviewer 1)

I have some questions on the percentages, lines 110-112

"77.8% of the participants to the study indicated that they had not been advised by a gynecologist on the harmful effects of smoking. This result differs between current smokers and non-smokers: 60.58% and 77.82%, respectively (p<0.001)."

If the results differ between 60.58% and 77.82% - first line "77.8% of the participants" must be incorrect. In all participants, this percentage must be somewhere between 60-77%, i.e. between smokers and non-smokers.

Thank you very much for this suggestion. There is indeed an error on this line. We investigated the results again and corrected the mistake (60.58% of 1266 smokers and 77.82% of 10182 non-smokers).

Line 110 "to the study" should be "in the study"

Thank you very much, we have changed these phrases as suggested by the reviewer.

Once again, thank you very much for your comments.

Yours faithfully, 

Authors

Reviewer 2 Report

REVIEW nr. 1380568

I checked the changes introduced by the authors, I have no particular requests to make, they seem properly addressed. I have seen the responses and modifications given to the first reviewer (particularly important with several criticalities). I suggest, if deemed appropriate, to have the review process carried out to date monitored by appointing a third reviewer.

Author Response

October 21th, 2021 

Dear Reviewers,

We would like to once again thank you very much for the remarks concerning our article entitled “Willingness to oppose smoking among pregnant women”. We marked all changes in the text in the track changes mode.

The answer to the second review report and changes made to the manuscript are presented below.

Review Report (Reviewer 2)

I checked the changes introduced by the authors, I have no particular requests to make, they seem properly addressed. I have seen the responses and modifications given to the first reviewer (particularly important with several criticalities). I suggest, if deemed appropriate, to have the review process carried out to date monitored by appointing a third reviewer.

Thank you very much for your review and the attention you paid to it. We are open to any guidance and will be enthusiastic to develop our manuscript if an appointment of a third reviewer is indicated.

Once again, thank you very much for your comments.

Yours faithfully, 

Authors
